# ATTENTION-BASED INTERPRETABILITY WITH CONCEPT TRANSFORMERS

**Mattia Rigotti, Christoph Miksovic, Ioana Giurgiu, Thomas Gschwind & Paolo Scotton**
IBM Research
Zurich, Switzerland
`{mrg,cmi,igi,thg,psc}@zurich.ibm.com`

## ABSTRACT

Attention is a mechanism that has been instrumental in driving remarkable performance gains of deep neural network models in a host of visual, NLP and multimodal tasks. One additional notable aspect of attention is that it conveniently exposes the "reasoning" behind each particular output generated by the model. Specifically, attention scores over input regions or intermediate features have been interpreted as a measure of the contribution of the attended element to the model inference. While the debate in regard to the interpretability of attention is still not settled, researchers have pointed out the existence of architectures and scenarios that afford a meaningful interpretation of the attention mechanism.

Here we propose the generalization of attention from low-level input features to high-level concepts as a mechanism to ensure the interpretability of attention scores within a given application domain. In particular, we design the Concept-Transformer, a deep learning module that exposes explanations of the output of a model in which it is embedded in terms of attention over user-defined high-level concepts. Such explanations are *plausible* (i.e. convincing to the human user) and *faithful* (i.e. truly reflective of the reasoning process of the model). Plausibility of such explanations is obtained by construction by training the attention heads to conform with known relations between inputs, concepts and outputs dictated by domain knowledge. Faithfulness is achieved by design by enforcing a linear relation between the transformer value vectors that represent the concepts and their contribution to the classification log-probabilities.

We validate our ConceptTransformer module on established explainability benchmarks and show how it can be used to infuse domain knowledge into classifiers to improve accuracy, and conversely to extract concept-based explanations of classification outputs. Code to reproduce our results is available at: `https://github.com/ibm/concept_transformer`.

## 1 INTRODUCTION

The spectacular gains in accuracy of recent large-scale machine learning models like deep neural networks have generally come at the cost of a loss of transparency into their functioning. This "black box" aspect severely limits their applicability in safety-critical domains, such as medical diagnostics, healthcare, public infrastructure safety, visual inspection for civil engineering, to name just a few, where it is essential for decisions to be corroborated by robust domain-relevant knowledge. In recent years, approaches focusing on explaining black box models have emerged, mostly with the goal of providing *post-hoc* explanations in terms of a set of relevant features used by the underlying model to make predictions (Ribeiro et al., 2016; Selvaraju et al., 2017; Lundberg & Lee, 2017; Smilkov et al., 2017). While widely used, such *post-hoc explainability methods* have been criticized for operating on low-level features such as pixel values, or sensory signals that are combined in unintelligible ways and do not correspond to high-level concepts that humans easily understand (Kim et al., 2018; Alvarez-Melis & Jaakkola, 2018; Kindermans et al., 2019; Su et al., 2019).

To overcome these limitations and sidestep the potential perils resulting from a misuse of post-hoc explainability of black box models, some researchers have been vocally advocating for the use

of *inherently interpretable models* (Rudin, 2019) that in particular would generate decisions based on human-understandable categories (i.e., concepts) grounded in domain expertise rather than raw features (Barbiero et al., 2021; Ghorbani et al., 2019; Kim et al., 2018; Yeh et al., 2020; Koh et al., 2020; Goyal et al., 2019; Kazhdan et al., 2020; Chen et al., 2020; Alvarez-Melis & Jaakkola, 2018; Li et al., 2018; Chen et al., 2019). For example, to identify a bird species, a model should focus on morphologically meaningful concepts, such as the shape, size and colors of beak, feathers or wings, rather than focusing on raw pixels, and combine them in ways that a domain expert (in this case an ornithologist) would reckon as intelligible to produce a classification. In addition, using high-level concepts emulates a human's thinking process (i.e., structured into familiar concepts) and provides insights into the model's reasoning in a human-understandable way.

The chasm between post-hoc explainability vs. inherently interpretable models closely reflects a related ongoing discussion in the NLP community on the interpretation of *attention mechanisms* (Bahdanau et al., 2014), and in particular on the interpretability of *attention weights* over input tokens, with researchers on one end of the debate claiming that attention provides interpretability, while others claim that "Attention is not explanation" (Jain & Wallace, 2019). While the debate over what degree of interpretability that can be ascribed to attention weights is still not settled (Wiegreffe & Pinter, 2019), it is arguable that in many situations attention is not a "fail-safe indicator" (Serrano & Smith, 2019), particularly when decisions rely on the interaction of multiple interacting tokens as is typically the case in deep architectures. Conversely then, a way to guarantee the interpretability of attention weights would be to make sure that they are not being processed by downstream operations that renders their relation to the decision outputs uninterpretable. This is indeed something that had been proposed in the past, in particular in architectures that preserve the interpretability of "relevance scores" (akin to attention weights) by acting on them only through a restricted class of intelligible "aggregation functions" such as additive models (Alvarez-Melis & Jaakkola, 2018), which are a common functional elements in interpretable and white-box models (Caruana et al., 2015).

In this paper, we propose the *ConceptTransformer (CT)*, a transformer-based module (Vaswani et al., 2017) for classification tasks, that can be used to enhance an arbitrary deep learning classifier with domain knowledge in the form of plausible cross-attention weights between input features and high-level interpretable concepts. The CT can be used as a drop-in replacement for the classifier head of any deep learning architecture. The resulting model can then be trained end-to-end without any additional overhead on the training pipeline, except a modification of the loss function that enforces plausibility of the explanation. Importantly, the CT was specifically conceived to provide explanations that guarantee *faithfulness by design* and *plausibility by construction*. Faithfulness is defined as the degree to which the explanation reflects the decision and aims therefore to ensure that the explanations are indeed explaining the model's operation (Lakkaraju et al., 2019; Guidotti et al., 2018). In our model this is achieved by enforcing a linear relation between the transformer value vectors that represent the concepts and their contribution to the classification log-probabilities. Plausibility refers to how convincing the interpretation is to humans (Guidotti et al., 2018; Carvalho et al., 2019). In the CT architecture, plausibility is achieved by construction by supervising the attention heads of the cross-attention mechanism to conform with inputs-concepts-outputs relations derived by domain knowledge.

We validate our approach on three image benchmark datasets, MNIST Even/Odd (Barbiero et al., 2021), CUB-200-2011 (Welinder et al., 2010), and aPY (Farhadi et al., 2009). On these datasets we will examine how the faithfulness and plausibility of the CT explanations are practically translated into domain-relevant explanations behind particular output decisions or diagnostic insights about ensuing wrong classifications. We will also quantify the benefit of domain-expert knowledge in terms of statistical efficiency by showing that providing domain-relevant explanations to our CT model tends to improve the performance of the downstream classification, in particular in low-data regime. This for instance translates in an 8-9% improvement in accuracy on the bird classification CUB-200-2011 dataset when CT is trained in conjunction with part location annotations.

We note in addition that one of the strengths of our CT model is its versatility that allows it to be effortlessly applied to other data modalities as well by combining it with deep learning classifiers which are then rendered interpretable with no appreciable overhead or change in their training pipeline. This is in stark contrast to other inherently interpretable models that are often specifically designed for the domain at hand, and require adhoc multi-stage training procedures. CT on the other hand, is differentiable and can be flexibly included in any end-to-end training pipeline that uses backpropagation, as we will showcase by combining it with a host of different deep learn-

ing backbones ranging from convolutional architectures like Residual Networks (He et al., 2016) to more modern Vision Transformer (Dosovitskiy et al., 2020) and hybrid Compact Convolutional Transformer models (Hassani et al., 2021).

## 2 RELATED WORK

In recent years, there have been significant advancements towards designing interpretable models that quantify the importance of individual features with respect to the prediction output. One general approach is post-hoc analysis, in which one interprets a trained model by fitting explanations to the classification outputs (Alvarez-Melis & Jaakkola, 2018; Ribeiro et al., 2016; Lundberg & Lee, 2017). In particular for CNNs, popular techniques are activation maximization (van den Oord et al., 2016; Nguyen et al., 2016; Yosinski et al., 2015) and saliency visualization (Selvaraju et al., 2017; Smilkov et al., 2017; Sundararajan et al., 2017).

However, these post-hoc methods do not actually explain how the underlying model reached a particular classification outcome. In contrast, attention-based interpretable techniques aim to expose which parts of a given input a network focuses on, and therefore deems to be most relevant, when making a decision. Examples of attention models are Zhang et al. (2014); Zhou et al. (2016; 2018); Zheng et al. (2017); Fu et al. (2017); Girshick (2015); Girshick et al. (2014); Huang et al. (2016). The problem with these models is that they focus on low-level individual features when providing an explanation. Such features are often not intuitive for humans, are typically noisy and non-robust, or can be misleading when interpreted afterwards (Kim et al., 2018).

One of the recent advancements in the field of interpretability was to design methods that explain predictions with high-level human understandable concepts (Ghorbani et al., 2019; Kim et al., 2018; Yeh et al., 2020; Koh et al., 2020; Goyal et al., 2019; Kazhdan et al., 2020; Chen et al., 2020; Barbiero et al., 2021; Li et al., 2018; Chen et al., 2019) – either by identifying common activation patterns in the last nodes of the neural network corresponding to human understandable categories or constraining the network to learn such concepts. For instance, TCAV (Kim et al., 2018) proposes to define concepts from user-annotated examples in which concepts appear. Others propose prototypes-based explanation models (Li et al., 2018; Chen et al., 2019), but they typically require specialized convolutional architectures to ensure feature extraction. In particular, ProtoPNet (Li et al., 2018) uses previously learned prototypes to focus attention on various parts of an image. This architectural design implies that object-level (global) concepts cannot be easily incorporated, and since prototypes are not learned together with the attention model, explanations based on these prototypes may lack faithfulness. SENN (Alvarez-Melis & Jaakkola, 2018) proposes a network that transforms inputs into interpretable basic features, generates concept relevance scores and then aggregates concepts with relevance scores to explain predictions. While it is out-of-the-box interpretable, it lacks concept localization. Barbiero et al. (2021) proposed a differentiable approach that allows the extraction of logic explanations from neural network models using First-Order Logic. The approach relies on an entropy-based criterion to automatically identify the most relevant concepts that have contributed to a particular classification output.

In our approach, high-level concepts are defined with a set of related dimensions, and can be part-specific or global. Such concepts are typically readily available in many domains and can be used to enhance the performance of the learning task while offering explainability at no additional cost for the network. Obtained explanations are plausible and guaranteed to be faithful, since concepts participate in the model computation. Finally, CT also allows in some cases to discover the presence concepts that were not annotated.

## 3 APPROACH

**The ConceptTransformer module.** The ConceptTransformer (CT) is a transformer-based module designed to be used as classifier head in a deep learning architecture that generates classification outputs using cross-attention (Vaswani et al., 2017) between input features and a set of embeddings representing domain-relevant concepts. Fig. 1 shows the case where the inputs to the CT are embeddings of $P$ visual patches of an input image that are linearly projected and concatenated into the query matrix $Q \in \mathbb{R}^{P \times d_m}$ of a query-key-value cross-attention mechanism whose corresponding key matrix $K \in \mathbb{R}^{C \times d_m}$ is the linearly projected concatenation of the embeddings representing

the $C$ concepts. In addition, the concepts are linearly projected with a value projection matrix and concatenated to result in the value matrix $V \in \mathbb{R}^{C \times d_m}$. Cross-attention then outputs an attention weight

$$\alpha_{pc} = \text{softmax}\left(\frac{1}{\sqrt{d_m}} Q K^\top\right)_{pc} \quad \text{with} \quad p = 1, \ldots, P, \quad c = 1, \ldots, C,$$

between each patch-concept pair, which are combined into an attention map matrix $A = [\alpha_{pc}] \in \mathbb{R}^{P \times C}$. The final output of the CT is the product obtained by multiplying the attention map $A$, the value matrix $V$ and an output matrix $O \in \mathbb{R}^{d_m \times n_c}$ that projects onto the (unnormalized) $n_c$ logits over the output classes, and averaging over patches:

$$logit_i = \frac{1}{P} \sum_{p=1}^{P} [AVO]_{pi} \quad \text{with } i = 1, \ldots, n_c. \tag{1}$$

Notice that here for simplicity we described a single-head attention model, but in our experiments we will be using a multi-head version (Vaswani et al., 2017).

Equation 1 says that, given an input $x$ to the network, the conditional probability of output class $i$ is

$$\Pr(i|x) = \text{softmax}_i \left(\sum_{c=1}^{C} \beta_c \, \gamma_c(x)\right) \quad \text{with } \beta_c \text{ with components } (\beta_c)_i = [VO]_{ci}, \tag{2}$$

and $\gamma_c(x)$ are *positive relevance scores* that depend on $x$ through the averaged attention weights: $\gamma_c(x) = \frac{1}{P} \sum_{p=1}^{P} \alpha_{pc}$. The output of the CT is therefore essentially a simple multinomial logistic regression model over positive variables $\gamma_c(x)$ that measures the contribution of each concept.

Notice that this result follows from the linear relation between the value vectors and the classification logits, which itself comes from the design choices of computing outputs from the value matrix $V$ through the linear projection $VO$, and aggregating patch contributions by averaging.

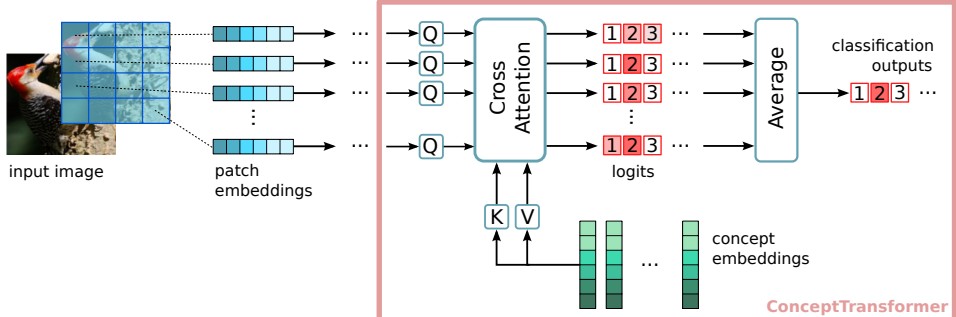

Figure 1: The ConceptTransformer (CT) is a transformer module that provides concept-based inter-pretability by design. It is a drop-in replacement for the classifier head of an arbitrary deep learning classifier and uses cross-attention to generate classification log-probabilities as additive contributions from user-defined concepts that are plausible in the domain under consideration. In the figure it is used as classifier head of a patch-based architecture like a ViT or a patch-level CNN.

**Faithful concept-based explanations by design.** The CT was conceived as a *drop-in replacement* for the classifier head of an arbitrary deep learning classifier to provide concept-based explanations of the outputs that are *guaranteed to be faithful by design*. We formalize this statement as follows:

**Proposition 1** *Each concept relevance score $\gamma_c(x)$ in Equation 2 is a* faithful *explanation of the output. More specifically, the probability of choosing the preferred output $i^c = \arg\max_i (\beta_c)_i$ of concept $c$ (assuming it's unique) is guaranteed to decrease if $\gamma_c(x)$ is externally set to zero. Moreover, the correlation between $\gamma_c(x)$ and $Pr(i^c|x)$ is strictly positive.*

**Proof** Proof of Proposition 1 is provided in Appendix A. ∎

Note that the last statement in the Proposition above is a corollary of the first one, and it specifically shows that CT is guaranteed to satisfy the technical definitions of *faithfulness* given for instance by Alvarez-Melis & Jaakkola (2018).

**Training and plausibility by construction.** As mentioned, the CT is a differentiable transformer-based module that can be embedded in a deep learning architecture trained end-to-end with back-propagation. In addition, the fact that it exposes attention weights over concept tokens that can be user defined gives us the freedom to shape these attention weights according domain-expert knowledge relevant for the problem under consideration. This can be done by explicitly guiding the attention heads to attend to concepts in the input that are a priori known to be informative to correctly classify the input. In practice this can be achieved by supervising the attention weights at training as for instance proposed by Deshpande & Narasimhan (2020) as a self-supervised technique for bi-directional language models. In particular, given a desired distribution of attention $H$ provided by domain knowledge (e.g., we know which patches in the input image contain which concepts that are relevant to classify the input) we can force the CT attention weights $A$ by adding an "explanation cost" term to the loss function that is proportional to $\mathcal{L}_{expl} = ||A - H||_F^2$, where $|| \cdot ||_F$ is the Frobenius norm. The final loss used to train the architecture then becomes $\mathcal{L} = \mathcal{L}_{cls} + \lambda \, \mathcal{L}_{expl}$, where $\mathcal{L}_{cls}$ denotes the original classification loss, $\mathcal{L}_{expl}$ the additional explanation loss, and the constant $\lambda \geq 0$ controls the relative contribution of the explanation loss to the total loss. Notice that setting $\lambda = 0$ essentially amounts to just minimizing the classification loss and disregarding the prior domain knowledge as imparted into CT by guiding the attention heads.

## 4 RESULTS

We have evaluated the proposed approach on three datasets: MNIST Even/Odd (mni), CUB-200-2011 (Welinder et al., 2010) and aPY (Attribute Pascal and Yahoo) (Farhadi et al., 2009). Each dataset illustrates a slightly different use case and different visual backbone in combination with the CT. This variety is meant to showcase the flexibility and versatility of the CT module.

With MNIST Even/Odd we consider a case where the correspondence between concepts and output classes is many-to-one and deterministic. In addition, concepts are global as opposed to being spatially localized. That means that we will not use a patch-based representation of the input, and instead the concepts refer to the whole input, which is essentially equivalent to having only one patch. For this task, CT is combined with a small Compact Convolutional Transformer architecture (Hassani et al., 2021).

In the case of CUB-200-2011 we consider a case where the relation between concepts and outputs is instead many-to-many and non-deterministic, and there exists a mixture of global and spatially localized concepts. In order to handle both types of concepts we instantiate two CTs, one for spatial concepts and one for global concepts and we then average the logits provided by both in order to preserve interpretability (as explained in the previous section). For this dataset, we use a Vision Transformer as a backbone. As inputs to the CT handling the spatial concepts we use the embeddings of the tokenized image patches, while as input to the CT in charge of the global concepts we use the embedding of the `CLS` token.

Finally, with the aPY (Attribute Pascal and Yahoo) dataset we consider another situation where concepts are exclusively global but this time are many-to-many. The architecture of the visual backbone used on this dataset is a larger version of the Compact Convolutional Transformer architecture used on MNIST Even/Odd, but with a larger tokenizer consisting of a ResNet50 model pre-trained on ImageNet.

### 4.1 EVALUATION ON MNIST EVEN/ODD

The objective of this evaluation is simply to illustrate how our architecture works. We perform a simple binary classification task, based on the MNIST Even/Odd dataset, aiming at classifying 28x28 pixel images of hand-written digits ranging from 0 to 9 as either 'even' or 'odd'. In this case, we exploit the fact that we know the identity of each digit and use that as an explanation for the binary classification prediction. In other words, for instance a '7' should be classified as 'odd', and a plausible explanation to support this prediction is that *it is 'odd' because it is a '7'*.

Figure 2 shows the accuracy on the test set (left) and explanation loss during validation (right), relative to the number of samples used at training, which varies from 100 to 7000. With the largest number of training samples, the CT achieves 99% accuracy, irrespective of whether concepts are leveraged ($\lambda = 2.0$) or not ($\lambda = 0.0$). However, when the model is trained with significantly

fewer samples (e.g., 100 - 500), using concepts provides a performance boost, in the range of 5%. Accordingly, we observe a more graceful decay of the explanation loss when $\lambda = 2.0$.

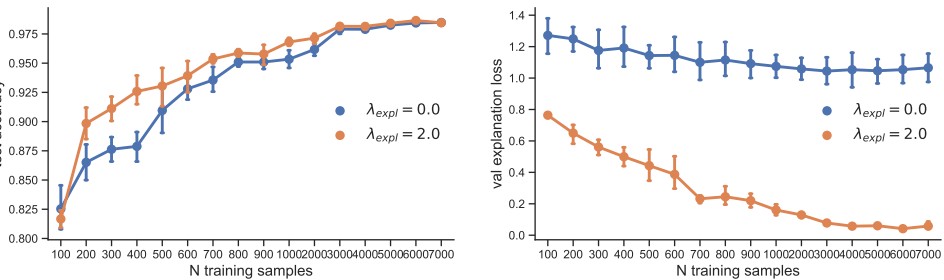

Figure 2: Sample-efficiency gain of using explanations at training on the MNIST Even/Odd dataset.

Figure 3 (left) shows a test sample, a '7' whose correct binary label is 'odd', which is itself a prediction that should be supported by the correct ground-truth explanation '7' (i.e., " *the sample is 'odd', because it is a '7'* "). CT in this case outputs the correct prediction, and looking at the concept attention weights we can see that indeed this prediction is supported by the correct concept.

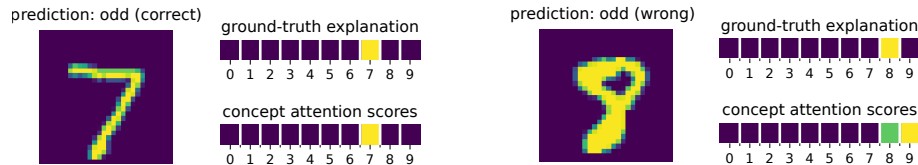

Figure 3: Examples of correct and incorrect predictions on MNIST Even/Odd.

Figure 3 (right) on the other hand shows the example of a test sample which is being misclassified by CT: the sample (an '8') is supposed to be classified as an even digit, but is instead classified as an odd number. If we look at the concept attention weights for this sample we see that the reason for the incorrect prediction can be traced back to the fact that the CT strongly associated the sample to the '9' concept and hence predicted the sample to be 'odd' because it "thought" that it was a '9'. Notice that also the correct concept '8' is being attended to by the architecture, but the wrong concept '9' received in this case a higher attention score. Interestingly, visually inspecting the sample it is not far from being a '9'.

## 4.2 EVALUATION ON CUB-200-2011

The CUB-200-2011 (Welinder et al., 2010) dataset contains 11788 images of birds, classified in 200 species. Each image is annotated with a given number of concepts (e.g., shape of the beak, color of the back, etc.) explaining the visual characteristics of the bird in the image. It uses 312 concepts, but their distribution varies across images. We therefore pre-process the dataset and retain concepts that are sufficiently representative in class-level annotations. Specifically, for a given class, we retain only concepts that are present in at least 45% of the images of that class and subsequently present in at least 9 classes. This led us to retain 108 concepts.

Table 1 compares our CT against other deep models (top models in bold), classified based on their training procedure in Multi-Stage (i.e., complex training) and End-to-end (i.e., training with back-propagation). When concepts are leveraged to train the CT, we notice a significant boost in accuracy of 8.2% (i.e., CT [w/o] vs. CT). Therefore, not only does the CT provide explainability by design, but it also boosts the overall performance of the classification task. In future, we plan to examine ways of modeling and taking advantage of the relation between extracted concepts. This has been shown to further boost performance by Barbiero et al. (2021) by extracting First Order Logic clauses that relate concepts.

The test accuracy of our model is on par with the baseline, non-interpretable model (B-CNN) as well as with ProtoPNet, PN-CNN, SPDA-CNN, MA-CNN and RA-CNN, even though some of these techniques require significantly more complex training than the CT. In terms of the type of

Table 1: Performance on CUB-200-2011 when concepts are leveraged at training (**CT**) or without concepts (**CT [w/o]**). Comparison against state-of-the-art approaches, classified by their training complexity: B-CNN (Lin et al., 2015b), Part R-CNN (Zhang et al., 2014), PS-CNN (Huang et al., 2016), PN-CNN (Branson et al., 2014), SPDA-CNN (Zhang et al., 2016), PA-CNN (Krause et al., 2015), MG-CNN (Wang et al., 2015), 2-level attn. (Xiao et al., 2015), FCAN (Liu et al., 2016), Neural const. (Simon & Rodner, 2015), ProtoPNet (Li et al., 2018), CAM (Zhou et al., 2016), DeepLAC (Lin et al., 2015a), ST-CNN (Jaderberg et al., 2015), MA-CNN (Zheng et al., 2017), RA-CNN (Fu et al., 2017). We report their best performance from (Li et al., 2018), irrespective of whether they are trained on full images or bounding boxes.

| Training | Accuracy [%] | | | |
|---|---|---|---|---|
| Multi-stage | Part R-CNN: 76.4 | PS-CNN: 76.2 | **PN-CNN: 85.4** | SPDA-CNN: 85.1 |
| | PA-CNN: 82.8 | MG-CNN: 83.0 | 2-level attn.: 77.9 | FCAN: 82.0 |
| | Neural const.: 81.0 | ProtoPNet: 84.8 | | |
| End-to-end | B-CNN: 85.1 | CAM: 70.5 | DeepLAC: 80.3 | ST-CNN: 84.1 |
| | MA-CNN: 86.5 | RA-CNN: 85.3 | CT [w/o]: 76.9$\pm$3 | **CT: 88.0$\pm$0.4** |

explanations provided, at the coarsest level, CAM highlights the entire object as the reason behind a classification decision. At a finer level, most models offer part-level attention, however they differ in how attention is generated. For example, Part R-CNN, SPDA-CNN, PS-CNN, PN-CNN use additional part-localization models previously trained with part annotations. RA-CNN uses an additional neural network to decide where to focus attention, whereas MA-CNN uses convolutional feature maps to direct attention to various parts of the image. ProtoPNet generates attention based on previously learned prototypes, namely it focuses on a specific region of the input image because the region is similar to a learned prototypical example. This leads to two main drawbacks. First, object-level concepts cannot be naturally incorporated, since the approach is based specifically on part-level concepts. Second, finding learned prototypes for other models requires post-hoc analysis, specifically inspecting which region mostly activates a model's convolutional filter. Since such prototypes are not computed together with the model, resulting explanations cannot be always guaranteed to be faithful to the classifier's decisions.

In contrast, the CT makes use of both part-level (*spatial*) and object-level (*global*) concepts to generate explanations, as shown in Fig. 4. We highlight concepts with the highest attention scores and hence most relevant for the classifier for four correctly predicted bird species. In addition, the model guarantees faithful explanations by learning concepts together with the classification model. Specifically, this is achieved by enforcing an additive relation between the transformer value vectors that represent the concepts and their contribution to the classification log-probabilities. We show that explanations mimic the reasoning of the classifier in Fig. 5 for one misdiagnosed bird species (left image). The CT predicts the species to be Summer Tanager, when in fact the bird is a Scarlet Tanager. If we look at the concept attention scores for this sample, we notice that there are two reasons for the incorrect prediction. First, relevant parts of the bird such as $upper\_parts\_color :: black$ and $has\_primary\_color :: black$ are not fully visible in the left most image. Both concepts are extremely important to ensure a correct classification as Scarlet Tanager, as seen in the middle image. Second, attention is not attended to the $has\_wing\_color :: black$ concept (missing), but instead to the $has\_shape :: perching - like$ concept, which is typical for the Summer Tanager species.

## 4.3 Evaluation on aPY (Attribute Pascal and Yahoo)

The aPY (Attribute Pascal and Yahoo) dataset introduced by Farhadi et al. (2009) consists of two subsets. The aPascal dataset was created for the Visual Object Classes Challenge 2008 by Everingham et al. and consists of 6340 training samples and 6355 test samples. aYahoo is a dataset developed by Farhadi et al. (2009) using Yahoo image search and comprises 2644 samples, which are typically used for testing only. Each sample image in either dataset contains one or more objects which are enclosed by a bounding box and labeled with a single class. In total there are 32 classes describing everyday objects (in the form of nouns) such as animals, vehicle types, or common items found in apartments. In addition, for most of the objects there is a set of associated attributes, denoting present object parts or characteristics, e.g., "tail", "hair", "door", or "screen". The number

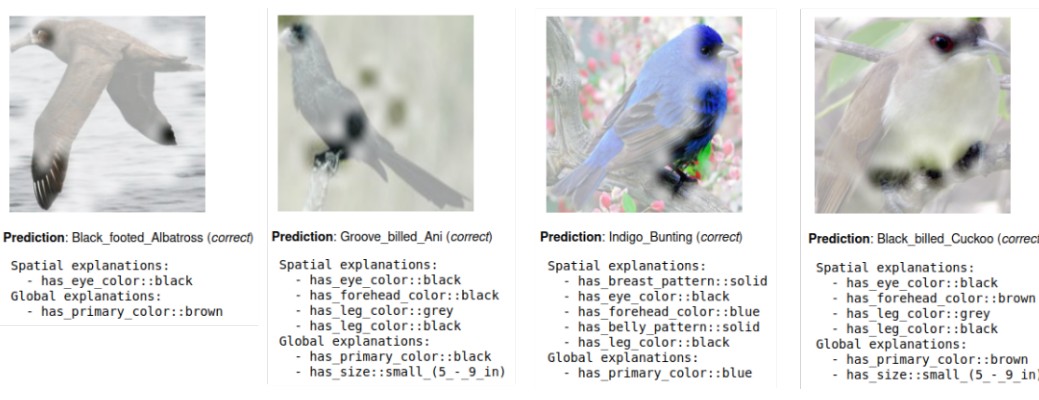

Figure 4: Attention scores for spatial (part-level) and global (object-level) concepts for four correctly predicted bird species from CUB-200-2011.

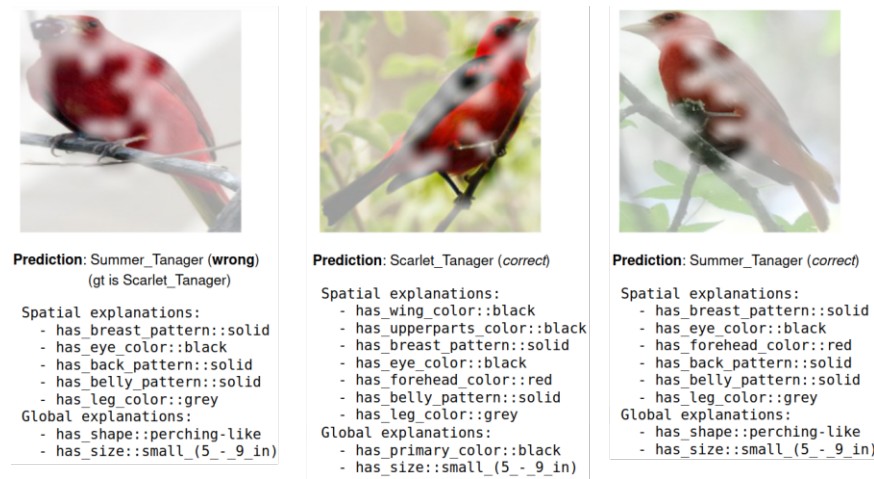

Figure 5: Diagnosing classification mistakes in CUB-200-2011.

of attributes is 64, with an average of around 20 present attributes per object while the minimum number of attributes is 2 and the maximum is 43. In contrast to the CUB dataset, there is no notion of object parts and associated attributes, i.e. the attributes are referring to the entire object ("global attributes"). Furthermore, the combined dataset of aPascal and aYahoo contains 989 objects without any attribute associated. We pruned these cases for our experiments so that 14350 objects remain. A cropped image was created for each bounding box in the samples of the dataset, yielding one image per object. At training time, the following augmentations were applied to the individual object samples: resizing to a standardized format ($H \times W = 320 \times 320$ pixels), random horizontal flipping with probability $p = 0.5$, random rotations in the range of $\pm 15°$ based on an uniform probability distribution and normalization.[1] For validation and testing, only resizing and normalization were applied.

We evaluated the aPascal as well as the combined aPY dataset with varying regularization parameter values applied to the explanation loss. Table 2 shows the negative log likelihood loss for the classification, the mean squared error (MSE) loss for the concepts (explanation loss) and accuracy figures for the aPascal and the aPY datasets. The accuracy numbers are in a narrow range, implying that the concept regularization does not seem to significantly affect the classification accuracy. On the other hand, the decrease of the concept-related MSE loss for increasing $\lambda$ values by approximately one order of magnitude suggests, that the labeled concepts are indeed learned by the model. The experiments were conducted with six different data-seeds and it was confirmed that the results are generalizable.

---

[1]We use the Albumentations library by Buslaev et al. (2020).

Table 2: Classification loss, explanation loss, accuracy on the PY test set and accuracy on the aPascal test set for selected concept regularization parameter values $\lambda$ after 500 epochs.

| $\lambda$ | NLL class | MSE concepts | Accuracy aPY [%] | Accuracy aPascal [%] |
|---|---|---|---|---|
| 0.0 | 0.1475 | 0.2463 | 61.2 | 84.7 |
| 0.5 | 0.1526 | 0.1152 | 60.1 | 85.1 |
| 1.0 | 0.1530 | 0.0855 | 61.3 | 85.6 |
| 2.0 | 0.1596 | 0.0737 | 61.5 | 85.8 |
| 5.0 | 0.1734 | 0.0596 | 61.2 | 85.4 |

Evaluations on the combined aPY dataset appear to be focused on specific applications like zero-shot learning and classifying attributes in the literature, which renders this dataset difficult to baseline. Still, Wang & Ji (2013) report 63.02% overall class prediction accuracy on the aPascal dataset, while (Mahajan et al., 2011) reached an accuracy of 67.33% on the same dataset.

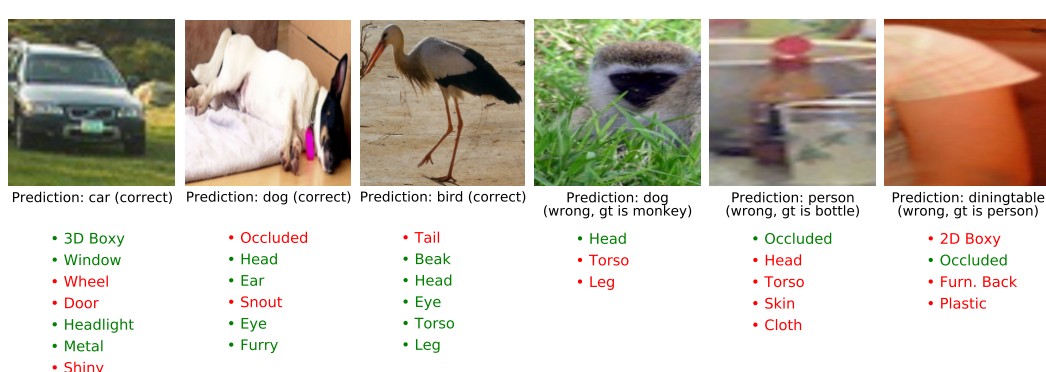

Figure 6: Correctly and incorrectly classified samples and activated global (object level) concepts above the given threshold from the aPY dataset. Red labels denote attributes that are not annotated in the ground truth.

Figure 6 shows samples of correctly and incorrectly classified objects and the activated concepts (i.e., above a given threshold) for the aPY dataset. Notably, our model is able to discover meaningful and obviously correct concepts that were not annotated in the ground truth, e.g., the "wheel", "door" and "shiny" attributes that pertain to the first image with the car, or the "snout" in the image with the dog. The incorrectly classified objects were associated with concepts that seem, to a certain extent, understandable. For example, the monkey (predicted as dog) or the bottle (predicted as person). In the latter case, we find that the classification is consistent with the concepts that were discovered, i.e., if "head", "torso", "arm", etc. are detected, it is coherent that the corresponding classification is "person".[2] It also shows that the concepts allow a person to better understand how the model arrived at the misclassification as well as more easily to identify it as such. We believe that the CT approach can be extended for applications in concept discovery and zero-shot learning.

## 5 CONCLUSIONS

In this work, we generalize the notion of attention from low-level features to high-level concepts to ensure an interpretability of attention scores that agrees with a human's reasoning for classification tasks. We propose the ConceptTransformer, a novel deep learning module that accommodates this form of interpretability and show on three benchmark image datasets that the explanations obtained are plausible and faithful. Our architecture achieves performance that matches the state-of-the-art techniques, while being more versatile, significantly less complex and easier to train.

---

[2]This resembles the role of attention weights in the NLP literature where they can be associated with syntactic elements in the input sentence.

## REPRODUCIBILITY

We provide code for the model implementation in the Appendix. Upon acceptance of the paper, we plan to release the full code for the experiments that we reported in a public repository. The datasets used for evaluation of our approach are public benchmarks, therefore already available in the community.

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

## A    Appendix: Additional details

Proof of Proposition 1

Let us fix a specific $c$ and let us denote $\Pr(i^c|x)$ in Equation 2 after we externally set $\gamma_c(x) > 0$ to zero with $\Pr(i^c|x)|_{\gamma_c=0}$.

We want to calculate the difference:

$$
\begin{aligned}
\Pr(i^c|x) - \Pr(i^c|x)|_{\gamma_c=0} &= \frac{\exp\left((\beta_c)_{i^c}\gamma_c + \sum_{c'\neq c}(\beta_{c'})_{i^c}\gamma_{c'}\right)}{\sum_i \exp\left((\beta_c)_i\gamma_c + \sum_{c'\neq c}(\beta_{c'})_i\gamma_{c'}\right)} - \frac{\exp\left(\sum_{c'\neq c}(\beta_{c'})_{i^c}\gamma_{c'}\right)}{\sum_i \exp\left(\sum_{c'\neq c}(\beta_{c'})_i\gamma_{c'}\right)} \\
&> \frac{\exp\left(\cancel{(\beta_c)_{i^c}\gamma_c} + \sum_{c'\neq c}(\beta_{c'})_{i^c}\gamma_{c'}\right)}{\sum_i \cancel{\exp\left((\beta_c)_{i^c}\gamma_c\right)}\exp\left(\sum_{c'\neq c}(\beta_{c'})_i\gamma_{c'}\right)} - \frac{\exp\left(\sum_{c'\neq c}(\beta_{c'})_{i^c}\gamma_{c'}\right)}{\sum_i \exp\left(\sum_{c'\neq c}(\beta_{c'})_i\gamma_{c'}\right)} \\
&= 0,
\end{aligned}
$$

where the inequality comes from the assumption that the preferred output $i^c = \arg\max_i(\beta_c)_i$ of concept $c$ is unique, i.e. $\exp((\beta_c)_{i^c}) > \exp((\beta_c)_i), \forall i \neq i^c$.

This proves that setting $c$ to zero causes a non-zero drop in $\Pr(i^c|x)$.

The same type of inequality more in general proves that increasing/decreasing $0 < \gamma_c(x) < 1$ increases/decreases $\Pr(i^c|x)$, which proves the second part of Proposition 1 that $\gamma_c(x)$ is strictly positively correlated with $\Pr(i^c|x)$.

## B    Appendix: Additional figures

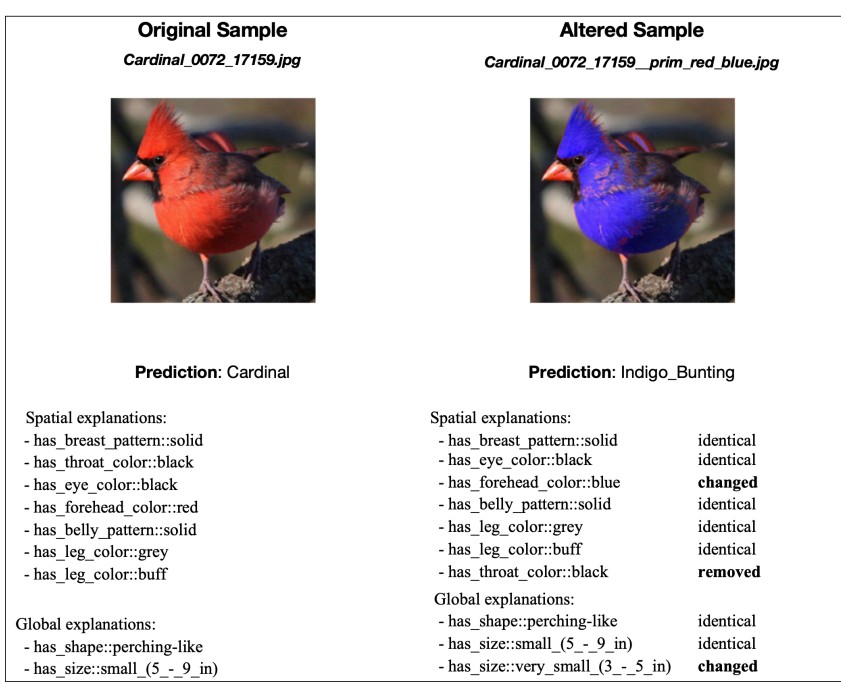

Figure 7: Counterfactual intervention on an exemplary CUB sample. Artificially changing the color of a Cardinal from red to blue, causes an attention shifts from the concept `has_forehead_color::red` to `has_forehead_color::blue` (among others). Correspondingly, the bird that was classified as a `Cardigan` is then being classified as an `Indigo Bunting`.

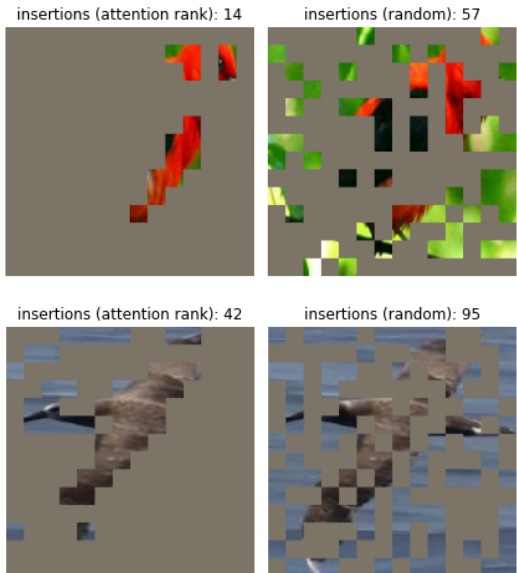

Figure 8: Illustration of the patch-based version of the insertion metric in Petsiuk et al. (2018). On the left patches corresponding to a sample in CUB are inserted one by one to the image according to the rank given by the maximal attention weight that they give to a concept until the classifier outputs the correct classification. On the right, the patches are inserted in random order. Inserting according to the rank of the attention weights allows the model to recognize the bird with less insertions, indicating that the attention weights are meaningfully related to the decision. This can be used as a measure of faithfulness captured by an insertion metric.

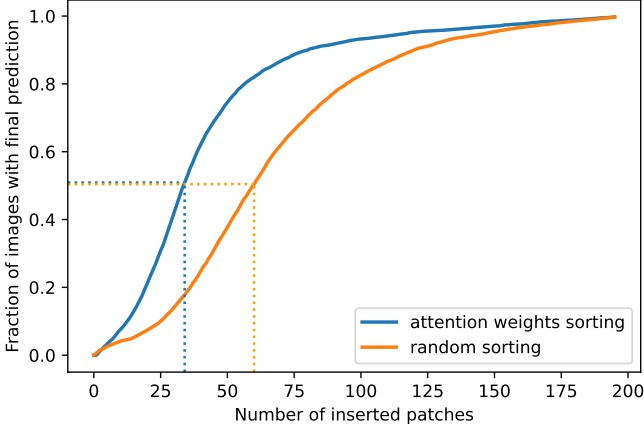

Figure 9: Patch-based insertion metric on CUB: inserting patches following the rank given by the attention scores learned by our ConceptTransformer an image has the probability of being recognized that reaches 50% of its maximum after 34 insertion. Random insertions on the other hand require on average 60 insertions.

## C  APPENDIX: CONCEPTTRANSFORMER PYTORCH CODE

```python
import math
import torch
import torch.nn as nn
import torch.nn.functional as F

class ConceptTransformer(nn.Module):
    """
    Processes spatial and non-spatial concepts in parallel and
    aggregates the log-probabilities at the end
    """

    def __init__(
        self,
        embedding_dim,
        num_classes,
        num_heads,
        attention_dropout=0.1,
        projection_dropout=0.1,
        n_concepts=10,
        n_spatial_concepts=0,
        *args,
        **kwargs,
    ):
        super().__init__()

        # Non-spatial concepts
        self.n_concepts = n_concepts
        self.concepts = nn.Parameter(torch.zeros(1, n_concepts, embedding_dim), requires_grad=True)
        nn.init.trunc_normal_(self.concepts, std=1.0 / math.sqrt(embedding_dim))
        if n_concepts > 0:
            self.concept_tranformer = CrossAttention(
                dim=embedding_dim,
                n_outputs=num_classes,
                num_heads=num_heads,
                attention_dropout=attention_dropout,
                projection_dropout=projection_dropout,
            )

        # Sequence pooling for non-spatial
        if n_concepts > 0:
            self.token_attention_pool = nn.Linear(embedding_dim, 1)

        # Spatial Concepts
        self.n_spatial_concepts = n_spatial_concepts
        self.spatial_concepts = nn.Parameter(
            torch.zeros(1, n_spatial_concepts, embedding_dim), requires_grad=True
        )
        nn.init.trunc_normal_(self.spatial_concepts, std=1.0 / math.sqrt(embedding_dim))
        if n_spatial_concepts > 0:
            self.spatial_concept_tranformer = CrossAttention(
                dim=embedding_dim,
                n_outputs=num_classes,
                num_heads=num_heads,
                attention_dropout=attention_dropout,
                projection_dropout=projection_dropout,
            )

    def forward(self, x):
        concept_attn, spatial_concept_attn = None, None

        out = 0
        if self.n_concepts > 0:
            token_attn = F.softmax(self.token_attention_pool(x), dim=1).transpose(-1, -2)
            x_pooled = torch.matmul(token_attn, x)

        if self.n_concepts > 0:  # Non-spatial stream
            out_n, concept_attn = self.concept_tranformer(x_pooled, self.concepts)
            concept_attn = concept_attn.mean(1)  # average over heads
            out = out + out_n.squeeze(1)  # squeeze token dimension

        if self.n_spatial_concepts > 0:  # Spatial stream
            out_s, spatial_concept_attn = self.spatial_concept_tranformer(x, self.spatial_concepts)
            spatial_concept_attn = spatial_concept_attn.mean(1)  # average over heads
            out = out + out_s.mean(1)  # pool tokens sequence

        return out, concept_attn, spatial_concept_attn
```

