# OpenReview forum: "Attention-based Interpretability with Concept Transformers"
_ICLR.cc/2022/Conference — ICLR 2022 Poster_

### Official Review · Reviewer_St45 · 2021-11-01

**Correctness:** 2
**Technical Novelty And Significance:** 2
**Empirical Novelty And Significance:** 1
**Recommendation:** 5
**Confidence:** 5

**Main Review:**

This work investigates the important field of interpretable deep models. With their technical contribution, the authors present an interesting approach to extend previous approaches, such as Bottleneck Concept Models (Koh et al.), by explicit local and global concepts.
The proposed approach seems to be very clearly described. Figure 1 and the illustrative example on MNIST describe the approach really well.

Furthermore, distinguishing between local and global concepts by the design of the architectures is a very interesting approach. Unfortunately, the paper only rudimentary focuses on this contribution. In this regard, the findings of the present work don’t show novel insights e.g. compared to Concept Bottleneck Models by Koh et al (ICML 2020) or the neuro-symbolic, object-centric approach by Stammer et al (CVPR 2021).

In particular, the takeaway message of Table 1 is not clear to me. In the main text, the authors mention that resulting explanations from ProtoPNet "cannot be always guaranteed to be faithful to the classifier’s decision", whereas the authors claim that the ConceptTransformer guarantees this, which is not clear to me. Are you only referring to ProtoPNet or Concept Bottleneck models in general? Can you elaborate on this further? Applying the evaluation of Margeloiu et al. (ICLR Workshop 2021) would further strengthen the authors’ claims.

The intention of the final evaluation on aPY is also not clear to me. Especially since here, only global concepts are used. What is the benefit compared to a more „simple“ classifier (c.f Concept Bottleneck Models by Koh et al.)?



References:

Koh et al. Concept Bottleneck Models, ICML 2020 (https://arxiv.org/abs/2007.04612)

Stammer et al. Right for the Right Concept: Revising Neuro-Symbolic Concepts by Interacting with their Explanations, CVPR 2021 (https://arxiv.org/abs/2011.12854)

Margeloiu et al. Do Concept Bottleneck Models Learn as Intended? ICLR Workshop 2021 (https://arxiv.org/abs/2105.04289)

**Summary Of The Paper:**

The paper introduces a transformer-based architecture for (deep) concept models. Similar to previous approaches in this area, the authors propose a replacement of the model’s deep classifier head. Here, the authors suggest using the attention mechanism. With the introduced ConceptTransformer module they are able to utilise global as well as local (spatial) concepts.
After introducing the architecture and supervised concept learning of the model, the model’s performance is demonstrated on three benchmark datasets. Where only one makes use of local and global concepts.

**Summary Of The Review:**

As it is now presented the paper seems only to be a minor technical extension of Concept Bottleneck Models and I would recommend that the authors focus more on presenting as well as evaluating the benefit of their ConceptTransformer.
The authors state that the explanations by their introduced approach are plausible and faithful. However, no quantitive evaluation in this regard is performed (cf. Margeloiu et al. Do Concept Bottleneck Models Learn as Intended?). The introduced ConceptTransformers seems to be a technical extension to „simple“ linear classification heads of Concept Bottleneck Models. The advantage is not clearly described and evaluated.
Therefore, the paper seems to me to be only a minor technical contribution.

---

> ### Author Response · Authors · 2021-11-22
> **Rebuttal to Reviewer St45 comments**
>
> We would like to thank the Reviewer the thoughtful comments and for pointing out related work that we will be happy to cite and thoroughly contrast with our own work in the camera-ready version of the paper.
>
> In principle, the Reviewer is right that, if one extends the definition of concepts from features to attention weights, then our model can be considered as a special type of Concept Bottleneck Model, where however the relation between concept activation and model output is mediated by cross-attention instead of the typical neural activation.
> This fact, together with the design choice of how attention scores for different input patches are being aggregated confers a few key properties that are not in general satisfied by a Concept Bottleneck model.
> For instance, the concept-based explanations provided by our ConceptTransformer in the form of the concept relevance scores $\gamma_c(x)$ are guaranteed to be faithful, in the following formal sense (that we now include in the rebuttal revisions of the paper):
>
> **Proposition 1** Each concept relevance score $\gamma_c(x)$ in Eq. (1) is a *faithful* explanation of the output.
> More specifically, the probability of choosing the preferred output $i^c=argmax_{i}(\beta_c)_i$ of concept $c$ (assuming it's unique) is guaranteed to decrease if $\gamma_c(x)$ is externally set to zero. Moreover, the correlation between $\gamma_c(x)$ and $Pr(i^c|x)$ is strictly positive.
>
> This result was only mentioned in words in the submitted version of the paper, but it is now spelled out as a Proposition together with a formal proof to emphasize that it is actually an important and formal result.
> We think that the fact that the ConceptTranformer's explanations are formally guaranteed to be faithful (i.e. they are by design contributing to the model output in an interpretable way), puts our contribution above the level of a mere incremental contribution to other concept-based explainability models.
> We will also be happy to cite the evaluation of Margeloiu et al. (ICLR Workshop 2021) in the final version of the paper and in particular note that our model is
> 1. "interpretable" in the sense used by Margeloiu et al., since unimportant concepts with $\gamma_c(x)=0$ are guaranteed not to contribute to the output $\Pr(i|x)=softmax_i\left(\sum\nolimits_{c=1}^C \beta_{c}~\gamma_c(x)\right)$
> 2. "predictable" in the sense used by Margeloiu et al., since the same expression above for our model's output indicates that the influence of the input $x$ is to the output is entirely mediated by the concepts $c$
> 3. "intervenable", because again from the same expression of the output we can see that increasing the concept relevance scores $\gamma_c(x)$ will increase the probability of the class $i^c=argmax_{i}(\beta_c)_i$.
>
> We hope this clarifies in what sense our concept-based explanations are guaranteed to be faithful and how for instance this contrasts with other models in Table 1 like ProtoPNet which do not provide this guarantee, and for which indeed faithfulness has been shown to be sometimes violated.
> As an addition to Table 1 we also would like to mention that we were now able to combine our ConceptTransformer with a larger Vision Transformer backbone to obtain a performance on CUB that is substantially higher than what we had reported in the submitted version of the paper. We now reach an accuracy on CUB of 88% which is substantially above the current state-of-the-art of explainability models (including ProtoPNet). This is reflected in Table 1 of Section 4.2 of the new version of the paper.
>
> Finally, the intentions behind the final evaluation on aPY were to demonstrate the versatility of the CT by (i) applying it to a scenario where the concepts would be global, as opposed to CUB where the concepts are both spatial and global, (ii) using a different visual backbones (ResNet50 instead of VisionTransformer), and (iii) testing a dataset and hence concepts from a broad domain (i.e.\ related to objects from everyday situations in aPY) in contrast to the datasets from CUB and MNIST, which are targeted to a narrower domain (birds in CUB, numbers in MNIST).

---

> > ### Comment · Reviewer_St45 · 2021-11-30
> > **Increase in score**
> >
> > Thank you for the response. The clarification on the differences to earlier Concept Bottleneck models addressed my main concern. I'm happy to raise my score to "marginally above the acceptance threshold".

---

> > > ### Author Response · Authors · 2021-11-30
> > > **Thank you very much**
> > >
> > > We are very glad that our clarifications were useful to address the Reviewer's comments, and grateful for the increase in score.
> > > We noticed that OpenReview still does not reflect the increase in score, and we wanted to make sure that this will be taken into account by the Area Chairs during the final evaluation of our paper. Thank you very much again.

---

### Official Review · Reviewer_H9Z1 · 2021-11-02

**Correctness:** 3
**Technical Novelty And Significance:** 3
**Empirical Novelty And Significance:** 3
**Recommendation:** 6
**Confidence:** 4

**Main Review:**

This paper presents a fairly straightforward yet novel modification to a vanilla Transformer that introduces concepts, which the user has to provide during training time. These concepts are then attended to via cross-attention against queries, which as usual come from the input. This allows an auxiliary, explanation loss to be used, which the authors show to be useful to increase classification accuracy. For most datasets presented, the authors' architecture is relatively competitive but does not achieve state of the art results, as I would expect from a model that baked in explainability. One cool bit is that classification accuracy does not necessarily fall with the use of the auxiliary loss (table 2), as one would expect it to.

A weakness of the work is that the concepts provide an explanation of perhaps why the model is wrong but what I expected was the ability to do some sort of fix of the model's attention so that it can be forced to output the right answer. As it is, I am not sure I am convinced that the concepts are causally linked to classification / misclassification performance. The authors provide a few examples where a misclassification happens and they show how the model attended to the wrong concepts, but is it possible this is a coincidence? The authors do not investigate in detail. It would be nice to see if the authors can change the attention weights in a natural way to explore this, such as for example altering the input image (ie, changing the color of certain pixels) so that certain concepts are / are no longer attended to and see how this affects classification performance.

**Summary Of The Paper:**

This work presents a new, Transformer-like architecture that has explainability built into it in the form of concepts which are then cross-attended against the input. The authors present their work in a variety of domains and show how it aids model decision interpretability.

**Summary Of The Review:**

A reasonable first draft of a promising line of work but more analysis is needed to make it interesting.

---

> ### Author Response · Authors · 2021-11-22
> **Rebuttal to Reviewer H9Z1 comments**
>
> We'd like to thank the Reviewer for taking time to review our work and providing suggestions for improving the paper.
> We carefully read the comments and we were able to carry out the additional experiments suggested by the Reviewer, which we hope will now satisfy all of the provided criticisms. These results have been added to the rebuttal revision of the paper.
> - In particular, the Reviewer noted as a weakness of our model that it didn't achieve state-of-the-art results. To address this, we now combined our ConceptTransformer with a larger Vision Transformer backbone and we were now able to achieve much better classification performance on CUB. Our new architecture now achieve 88% accuracy (Table 1, Section 4.2 in the paper), which is now the state-of-the-art performance by a decisive margin among explainable models on this dataset.
> - We now elaborate on the faithfulness guarantee of the concept-based explanations provided by ConceptTransformer. These were just mentioned in the text, but we now rewrote them in a more formal way to clarify exactly the points pointed out by the Reviewer. In particular, we state and prove the following Proposition regarding our ConceptTranformer (copied from the rebuttal revision of the paper):
>
> -  **Proposition 1** Each concept relevance score $\gamma_c(x)$ in Eq. (1) is a *faithful* explanation of the output.
> More specifically, the probability of choosing the preferred output $i^c=argmax_{i}(\beta_c)_i$ of concept $c$ (assuming it's unique) is guaranteed to decrease if $\gamma_c(x)$ is externally set to zero. Moreover, the correlation between $\gamma_c(x)$ and $Pr(i^c|x)$ is strictly positive.
>
> - As asked by the Reviewer, this proposition demonstrates that the concept relevance scores $\gamma_c(x)$ are "causally" linked to the output, in the sense that counterfactually setting them to zero will decrease the probability of the output associated to class $argmax_{i}(\beta_c)_i$, which also confirms that the link between attending to the wrong concept and a misclassification is not coincidental as suggested by the Reviewer. In our ConceptTransformer model this link is guaranteed to be "causal" by design.
> Conversely, controlling a concept relevance score by increasing it externally (or through an external module), is guaranteed to force the output to decide in favour of the preferred class of the concept.
> This shows that our model's explanations can indeed be intervened upon to correct an output, which addresses another weakness indicated by the Reviewer.
> - Finally, we performed the alteration experiment suggested by the Reviewer and changed the color of a red Cardinal in CUB to blue, again to showcase the faithfulness of the explanations provided by our model. As result the model changed its attention from the concept has_forehead_color::red to the concept concept has_forehead_color::blue. In correspondence to this switch in attention, the model also recognized the modified Cardinal as an Indigo bunting, which indeed is a blue bird (plots are included in Appendix B of the revision version of the paper).
>
> We again thank the Reviewer for the suggestions for improvements, and we hope that the ways we incorporated them in the rebuttal revision of the paper will be satisfactory and enough to justify an increase in our acceptance score.

---

### Official Review · Reviewer_wR5k · 2021-11-02

**Correctness:** 4
**Technical Novelty And Significance:** 2
**Empirical Novelty And Significance:** 2
**Recommendation:** 5
**Confidence:** 4

**Main Review:**

Quality and clarity:
The quality of the paper is good and it has a good motivation. The targeted goal is interesting and meaningful and the authors achieved this by linearly enforcing domain knowledge to weight allocations of attention mechanism. The proposed methodology is well-described over the paper and introduced experimental settings are properly designed with three public datasets. The written
results seem to be reasonable.

Originality:
Although interesting, It’s a bit hard to say the introduced technique is quite new, since a simple linear combination of attention weights and concepts patches (domain knowledge) in Transformer is the main contribution. Also, the experimental results are relatively weak when comparing the main model with its baselines in accuracy performance.


Qualitative and quantitative analysis of model explainability between the proposed model and its baselines can be a good validation to show the benefit of their explanation technique. The authors also need to describe how concept patches are generated and the error bound generated from the concept patches should be considered for clarity.

**Summary Of The Paper:**

The authors propose a transformer-based model which can enhance explainability of a deep learning model by inducing domain knowledge into the model in a form of cross-attention mechanism. This paper interestingly focuses on addressing the relationship between post-hoc  explainability and inherent  interpretable model.  The authors address the limitation of interpretable model, which focuses on the controversy over how much a human should trust model’s explanations for its decisions.


**Summary Of The Review:**

Although the paper is interesting, it's considered that the proposed technique is somewhat incremental.

---

> ### Author Response · Authors · 2021-11-22
> **Rebuttal to Reviewer wR5k comments**
>
> Thank you very much for taking the time to review our paper in depth.
> We carried out some additional experiments and planned a series of modifications to the paper to address the Reviewer's comments in detail.
> In particular, we were able to address (we hope satisfactorily) the criticism that our model displays relatively weak accuracy compared to the baseline models by combining the ConceptTransformer with a larger VisionTransformer model which now allows us to reach an accuracy of 88% on CUB (Table 1, Section 4.2). This is largely above the state-of-the-art performance for interpretable models.
>
> We also clarify that the design choices that lead to the linear relation between the value embeddings and the output of the ConceptTransformer are actually crucial to provide the theoretical guarantees that our model's concept-based explanations are faithful (in the technical sense proposed e.g. by Alvarez-Melis and Jaakkola, 2018 that relevance scores are positively correlated with output probabilities).
> We clarify this points in the revised version of the paper by formulating and proving the following proposition (that was already in the submitted paper but only presented informally):
>
> **Proposition 1** Each concept relevance score $\gamma_c(x)$ in Eq. (1) is a *faithful* explanation of the output.
> More specifically, the probability of choosing the preferred output $i^c=argmax_{i}(\beta_c)_i$ of concept $c$ (assuming it's unique) is guaranteed to decrease if $\gamma_c(x)$ is externally set to zero. Moreover, the correlation between $\gamma_c(x)$ and $Pr(i^c|x)$ is strictly positive.
>
>
> As far as we know, our model is the first that combines the flexibility of being able to be used with virtually any deep learning classifier, with the strong theoretical guarantees in terms of faithfulness of the explanations that it provides.
> Explainability methods in deep learning in fact tend not to provide any guarantees in terms of the faithfulness of their explanations. In fact, this is one of the main criticisms that are raised on them and one of the main weaknesses compared to inherently interpretable models.
>
> In the revised version of the paper we also show in practice what the faithfulness guarantees entail in practice. For this we already generated a plot that should answer directly the question of the Reviewer to connect the attention weights from individual patches to the output of the model. In particular, we computed an insertion score to quantify the faithfulness of patch-based explanations on CUB as done in the paper Petsiuk & Das & Saenko, BMVC (2018). Specifically, starting from an empty image, we inserted patches one by one starting from those with highest attention weights and monitored the evolution of the probability of recognizing the bird. We saw that the classifier was able to recognize the correct bird with a probability above 50% of that of the full image, already with ~34 of the 196 patches. On the other hand, if the patches were inserted at random, it took almost twice as many patches to reach the same probability. This analysis can be found in Appendix B of the rebuttal version of the paper.
> We also performed targeted counterfactual interventions on concepts by for instance artificially changing the color of birds, which reveled that, as one could plausibly expect, changing the color of a Cardinal from red to blue would cause its forehead to be attributed the blue color, and correspondingly the whole bird to be recognized an an Indigo Bunting (this analysis is also included in Appendix B).
>
> We hope that these additions to the revised version of the paper will meaningfully address the Reviewer's comment, and in particular convince them that our work is not merely an incremental iteration over the typical post-hoc explainability approach in deep learning, but a promising step in trying to bridge the gap between high-performance deep learning architectures and inherently interpretable models, as the Reviewer also alluded to at the beginning of the review. In this sense, we'd like to argue that our work is a conceptual departure that is trying to reconcile methodologies that are usually perceived as contradictory like deep learning and inherent interpretability.

---

### Official Review · Reviewer_5kGA · 2021-11-06

**Correctness:** 4
**Technical Novelty And Significance:** 3
**Empirical Novelty And Significance:** 3
**Recommendation:** 8
**Confidence:** 4

**Details Of Ethics Concerns:**

I don't see ethical concerns.

**Main Review:**

I think the paper explores an important problem and propose an interesting model that goes in a good direction in terms of encouraging models to not just achieve stronger numerical performance but also perform explainable reasoning along the way to justify their predictions.
* **Writing quality**: The writing clarity of the paper is good, the idea and content are easy to follow, the motivation is clearly explained, and there are good visualizations that help understanding.
* **Related work**: the paper misses some closely related prior works that explored the idea of reasoning over concepts for reasoning over images: one paper is “The consciousness prior” by Bengio and the other is “Learning by Abstraction: the Neural State Machine” by Hudson and Manning. The former discussed the idea mainly in the conceptual level and the latter studied it also empirically. At the same time, overall the related work section gives a good coverage of prior works about interpretability and attention.
* **Model structure**: the model proposed is simple yet interesting and general and so can be easily incorporated into transformer-based models for variety tasks and in different domains.
* **Attention supervision**: I really like the idea of supervising the concept attention directly when such information exists, as in the case of CUB. I think that’s a nice advantage of the proposed approach.
* **Experiments**: The experimental section is overall extensive enough and provides good balance both over multiple datasets, some with extra supervision info (like CUBs) and some more diagnostic (like MNIST), experiments include both quantitative and qualitative results. To demonstrate the potential generality of the approach, it could be nice to explore over different domains (e.g. textual only) or tasks (e.g. VQA, or other multimodal tasks) but I feel in the context of the proposed model exploring just image classification is nevertheless alright and not too limited.
* **Concepts identification**: one potential limitation of the approach proposed is the fact that the concepts needed to be pre-defined rather than emergent through the model, which may conversely reduce the applicability or potential benefit of the approach in all cases where such info doesn’t exist.


**Summary Of The Paper:**

The paper proposes a new version of transformer called concept transformer with the aim of improving the interpretability of its attention by computing cross attention between the inputs features and a set of concepts. This should make the model both more explainable, in the sense that it is easier for the human to interpret the attention weights of meaningful concepts, and also more faithful, in the sense that the attention scores given to particular concepts directly impact the final prediction of the model.

Update: Following the author response I'd like to keep my positive score reflecting my view of the paper.

**Summary Of The Review:**

Overall, I think the paper proposed a nice interesting model on an important problem, motivates the goal of the model in a compelling manner, presents the model with good detail and clarity, provides a nice mix of experiments of different types and on different datasets, and can be of benefit to the research community, and therefore I recommend its acceptance and wish best of luck to the authors.

---

> ### Author Response · Authors · 2021-11-22
> **Rebuttal to Reviewer 5kGA comments**
>
> We would like to thank the Reviewer for the attentive review and for the kind words of encouragement.
> Our model does indeed strive to help making the predictions of an (arbitrary) deep learning model human-interpretable, and doing so without getting in the way of performance.
>
> We will be happy to cite the papers mentioned by the Reviewer in the camera-ready version of the paper. Even though they are not specific to explainability and interpretability, they give a good conceptual background on the interpretation and mechanistic importance of attention weights.
>
> In our model we see the fact that concepts are pre-defined rather than emerging from end-to-end training as a design choice stemming from our desire to provide *plausible* explanations, i.e. explanations that are grounded in the domain knowledge of the practitioner making use of the model. Our ConceptTransformer could in fact as well be trained end-to-end without supervising the explanations, and the resulting explanations would still be guaranteed to be faithful (in the way that we now technically specify of being "causally" related to the output of the model). However, whether this explanations are also plausible for domain experts wouldn't be guaranteed and would have to be verified a posteriori after training.
> By offering the possibility to supervise the explanations, our model on the other hand gives control to the domain experts over the factors that they want to be driving the output, so that they can ensure that the model's decisions are based on factors are interpretable and/or mechanistically relevant.
> In future work we in any case plan to investigate what type of explanations are discovered by our ConceptTransformer if they are not supervised, as suggested by the Reviewer.

---

### Decision · Program_Chairs · 2022-01-20

**Decision:**

Accept (Poster)

**Comment:**

This paper proposes a generalization of the standard Transformer attention mechanism in which keys and queries represent abstract concepts (which must be specified a priori). This in turn yields "concept embeddings" (and logits) as intermediate network outputs, providing a sort of interpretability. Reviewers agreed that this is a simple (in a good way) and interesting approach, and may lead to follow-up work that builds on this architecture.

Some concerns regarding the relation of this method to prior work—in particular the "Concept Bottleneck" model—were raised and addressed in discussion; the authors might incorporate additional discussion in future drafts of the work.